

# Canopy uptake dominates nighttime carbonyl sulfide fluxes in a boreal forest

Linda M.J. Kooijmans[1], Kadmiel Maseyk[2], Ulli Seibt[3], Wu Sun[3], Timo Vesala[4,5], Ivan Mammarella[4], Pasi Kolari[4], Juho Aalto[4,6], Alessandro Franchin[4,8], Roberta Vecchi[7], Gianluigi Valli[7], Huilin Chen[1,8]

[1]Centre for Isotope Research, University of Groningen, Groningen, The Netherlands.
[2]School of Environment, Earth and Ecosystem Sciences, The Open University, Milton Keynes, United Kingdom.
[3]Department of Atmospheric and Oceanic Sciences, University of California, Los Angeles, California, USA.
[4]Department of Physics, University of Helsinki, Helsinki, Finland.
[5]Department of Forest Sciences, University of Helsinki, Helsinki, Finland.
[6]SMEAR II, Hyytiälä Forestry Field Station, University of Helsinki, Korkeakoski, Finland.
[7]Department of Physics, University of Milan, and INFN, Milan, Italy.
[8]Cooperative Institute for Research in Environmental Sciences (CIRES), University of Colorado, Boulder, Colorado, USA.

*Correspondence to*: Huilin Chen (huilin.chen@rug.nl)

**Abstract.** Nighttime vegetative uptake of carbonyl sulfide (COS) can exist due to the incomplete closure of stomata and the light-independence of the enzyme carbonic anhydrase, which complicates the use of COS as a tracer for gross primary productivity (GPP). In this study we derived nighttime COS fluxes in a boreal forest (the SMEAR II station in Hyytiälä, Finland; 61°51′ N, 24°17′ E, 181 m ASL) from June to November 2015 using two different methods: eddy-covariance (EC) measurements ($F_{COS-EC}$) and the radon-tracer method ($F_{COS-Rn}$). The nighttime COS fluxes averaged over the whole measurement period were -8.1 ± 1.5 and -7.9 ± 3.8 pmol m$^{-2}$ s$^{-1}$ for $F_{COS-Rn}$ and $F_{COS-EC}$, respectively, which is 38 % of the average daytime fluxes and 21 % of the total daily COS uptake. The correlation of $^{222}$Radon (of which the source is the soil) with COS (average $R^2$ = 0.59) was lower than with $CO_2$ (0.79), suggesting that the main sink of COS is not located at the ground. These observations are supported by soil chamber measurements that show that soil contributes to only 33 % of the total nighttime COS uptake. We found a decrease of COS uptake with decreasing night-time stomatal conductance and increasing VPD and air temperature, driven by stomatal closure in response to a warm and dry period in August. We also discuss the effect that canopy layer mixing can have on the radon-tracer method and the sensitivity of $F_{COS-EC}$ to atmospheric turbulence. Our results suggest that the nighttime uptake of COS is mainly driven by the tree foliage and is significant in a boreal forest, such that it needs to be taken into account when using COS as a tracer for GPP.

## 1 Introduction

The global budget of carbonyl sulfide (COS) is of interest for both stratospheric and tropospheric chemistry (Watts, 2000; Kettle et al., 2002, Berry et al., 2013, Launois et al., 2015). COS contributes to the formation of the sulfate aerosol layer in the stratosphere (Crutzen, 1976; Chin and Davis, 1995) and thereby also plays a role in ozone depletion (Brühl et al., 2012). In the troposphere COS is linked to the carbon cycle because it follows the same diffusion pathway into plant stomata as $CO_2$





during photosynthesis. After COS has entered a plant cell it is hydrolyzed by the enzyme carbonic anhydrase (CA) to form $H_2S$ and $CO_2$ (Protoschill-Krebs and Kesselmeier, 1996). As this reaction is practically irreversible, COS is not re-emitted by plants, in contrast to $CO_2$. The close coupling of COS and $CO_2$ uptake fluxes by vegetation makes COS a potentially powerful tracer for estimates of gross primary production (GPP; Sandoval-Soto et al., 2005; Montzka et al., 2007; Campbell et al., 2008; Wohlfahrt et al., 2012; Asaf et al., 2013).

Besides the difference in re-emission, the COS and $CO_2$ uptake processes differ in the sense that the consumption of COS by the CA enzyme is light-independent. This means that vegetative uptake of COS can continue during the night if stomata are not completely closed (Maseyk et al., 2014). Caird et al. (2007) showed that nighttime stomatal conductance exists in a wide variety of plant species and several studies report nighttime depletion of COS mole fractions (White et al., 2010; Belviso et al., 2013; Commane et al., 2013; Berkelhammer et al., 2014; Billesbach et al, 2014; Maseyk et al., 2014; Commane et al., 2015; Wehr et al., 2017). The measurements presented in White et al. (2010), Maseyk et al. (2014), Berkelhammer et al. (2014) and Wehr et al. (2017) indicated that nighttime ecosystem COS fluxes were indeed dominated by the vegetation, and not by the soil. In these studies, nighttime vegetative fluxes varied between 25 and 50 % of average daytime fluxes. A correlation between nighttime COS fluxes and stomatal conductance is expected when the nighttime sink of COS is primarily driven by vegetative uptake. The relation between $H_2O$ and COS fluxes shown by Seibt et al. (2010), Wohlfahrt et al. (2012) and Berkelhammer et al. (2014) underpins the likely relation between stomatal conductance and COS fluxes. However, the relation between COS fluxes and stomatal conductance measurements has not been studied under field conditions. Instead, Wehr et al. (2017) used COS ecosystem fluxes to estimate stomatal conductance. This relation can especially be useful for estimating nighttime stomatal conductance, which cannot be accurately determined under humid conditions as the concentration gradient of water vapor in leaf chambers gets too small (Maseyk et al., 2014).

Although COS is not used as a GPP tracer during nighttime conditions (when GPP is zero), nighttime COS fluxes may interfere with the use of COS for GPP estimates (Berry et al., 2013; Berkelhammer et al., 2014). To analyze the role of nighttime COS fluxes on the total COS budget and study correlations with environmental drivers, it is key to determine nighttime COS fluxes accurately. Eddy-covariance (EC) is a well-established technique to determine ecosystem fluxes (Aubinet et al., 2012); however, stable nighttime conditions complicate the measurements due to non-turbulent processes like canopy-layer storage and advection (Papale et al., 2006; Wohlfahrt et al., 2012; Aubinet et al., 2012). A method that has been used to derive specifically nighttime fluxes of trace gases, including COS, is the radon-tracer method (Schmidt et al., 1996; Van der Laan et al., 2009; Belviso et al., 2013). This method relates the nighttime buildup of trace gas concentrations to that of $^{222}$Radon ($^{222}$Rn) concentrations and the $^{222}$Rn flux, which is solely driven by the soil. Both the EC and radon-tracer methods can complement each other to help understand and reduce uncertainties of nighttime flux measurements.





The aim of this study is to quantify nighttime COS fluxes to determine the role of these fluxes in the ecosystem COS budget, and to understand the driving parameters of nighttime COS uptake. In the summer of 2015, we conducted a field campaign in a Finnish boreal forest using a combination of COS measurements: atmospheric concentration profiles, and EC and soil chamber measurements. We use both the EC and radon-based fluxes to quantify nighttime COS fluxes and infer information

about the sink apportionment within the canopy. We also investigate the correlation of nighttime COS fluxes with stomatal conductance and environmental parameters and discuss the implications of nighttime COS fluxes for large-scale GPP estimates.

## 2. Field measurements and data

### 2.1 Measurement site

The field campaign was held from June to November 2015 at the Station for Measuring Forest Ecosystem-Atmosphere Relations (SMEAR II) in Hyytiälä, Finland (61°51′ N, 24°17′ E, 181 m ASL). The forest represents boreal coniferous forest and the measurement site is covered by 50-60 year old Scots pine (*Pinus sylvestris*) up to 1 km towards the North from the measurement site and for about 200 m in all other directions (Rannik, 1998; Rannik et al., 2004). The forest outside this area covers younger pine and spruce. About 700 m southwest of the measurement site is an oblong lake of about 200 m wide. The

dominant canopy height is 17 m and the site is characterized by modest height variation. At this latitude, the daylight duration has a maximum in June with 19 hours and 40 minutes and is 7 hours in November.

### 2.2 Instrumentation for measurements of COS, $CO_2$, and $H_2O$.

Two quantum cascade laser spectrometers (QCLS) manufactured by Aerodyne Research Inc. (Billerica, MA, USA) were deployed in the field for simultaneous measurements of COS, $CO_2$, CO, and $H_2O$ and are described separately in the

following two sections.

### 2.2.1 QCLS for vertical profile and soil flux measurements

From June 1 until November 4, one QCLS was operated at 1 Hz for concentration measurements of sampled air at 4 heights: 125 m (tall tower), 23 m, 14 m, and 4 m (small tower at 30 m distance from the tall tower). An additional height of 0.5 m was measured as part of the soil chamber measurement routine from June 28 onwards. A multi-position Valco valve (VICI;

Valco Instruments Co. Inc.) was used to switch between the sample tubing from the different profile heights, soil chambers and calibration cylinder gases. The following measurements were made during each hour: 3 minutes for each of the four heights, 16 minutes for each of the two soil chambers, two times 3 minutes for one calibration cylinder to correct for instrument drift, 3 minutes for each of two other calibration cylinders to assess the accuracy of the measurements. The three cylinders were filled with ambient air and calibrated against two NOAA/ESRL standards for COS (NOAA-2004 scale) and

$CO_2$ (WMO-X2007 $CO_2$ scale) at the University of Groningen. A 'zero' air spectrum was measured once every six hours





using high-purity nitrogen (N 5.0). The overall uncertainty including scale transfer, water vapor corrections, and measurement precision of this analyzer was determined to be 7.5 ppt for COS and 0.23 ppm for $CO_2$ (Kooijmans et al., 2016). More detailed information about the calibration and correction methods can be found in Kooijmans et al. (2016).

### 2.2.2 QCLS for eddy covariance measurements

A second QCLS was used to measure COS, $CO_2$, CO, and $H_2O$ concentrations at 10 Hz from June 28 onwards. The air is sampled at 23 m height at a small tower that is at 30 m distance from the 125 m tall tower. Wind velocity components were measured by a sonic anemometer (Solent Research HS1199, Gill Ltd., Lymington, Hampshire, England) to derive ecosystem fluxes through the EC method. For this analyzer a 'zero' air spectrum was measured once every 30 minutes. This QCLS was calibrated against a standard on the same scale as the first QCLS. The $CO_2$ and $H_2O$ fluxes from the QCLS were compared

with those obtained at the nearby tall tower as quality control. The instrumentation in the tall tower is a Gill Solent 1012R anemometer and a Li-Cor LI-6262 gas analyzer (Mammarella et al., 2009).

### 2.3 Soil chambers

Two soil flux chambers (LI8100-104C; Li-Cor) modified for analysis of COS were used in combination with the concentration measurements of the QCLS at 1 Hz to derive soil fluxes. The modifications included operation in an open flow

configuration, replacing the chamber bowl and soil collar with stainless steel components, and removing or replacing other COS-producing material. Each chamber was closed once per hour for 9 or 10 minutes. For supply flow into the chambers, air was sampled at 0.5 m height in the vicinity of the soil chambers and was measured for 3 minutes before and after chamber closure. The air was pumped into the chambers with flow rates between 1.5 and 2.1 L min$^{-1}$ through a diaphragm pump (KNF 811) for which we found no interference with COS. More details on the soil measurements can be found in Sun et al.

20   (2017).

### 2.4 Auxiliary data

### 2.4.1 $^{222}$Radon

$^{222}$Rn concentrations were obtained by measurement of its short-lived decay products attached to aerosol particles (i.e. $^{214}$Bi). Detection of short lived decay products concentration in outdoor air was done by continuous on-line alpha spectroscopy

during aerosol sampling. Aerosol particles were collected at 8 m height as part of the ongoing aerosol monitoring at the site (Hari and Kulmala, 2005; Nieminen et al., 2014) about 50 m away from the tower where COS and $CO_2$ was sampled. Particles were collected on a glass micro fibre filter (Whatman GF/A, 47-mm diameter) with an average flowrate of 17.4 l min$^{-1}$. Alpha particles emitted by Radon decay products were recorded by a silicon surface barrier detector (ULTRA™ Alpha Detector by ORTEC, with F.W.H.M. of 42 keV) placed a few millimeters in front of the filter in order to optimize the

efficiency and to allow the detection of alpha particles in air. The hourly alpha energy spectra were continuously recorded.





The concentration of radon daughters is calculated by taking into account radioactive decay equations, the accumulation of decay products on the filter during the sampling and the hypothesis of equilibrium in the progeny after subtraction of the $^{220}$Radon daughter contribution. Following Schmidt et al. (1996), $^{222}$Rn and its decay products were considered in secular radioactive equilibrium in this work. Further details on the experimental procedure are reported in Marcazzan et al. (2003)

and Sesana et al. (2003).

### 2.4.2 Stomatal conductance

The stomatal conductance to water vapor ($g_{sw}$) was determined from transpiration measurements obtained through shoot chamber measurements at a pine shoot at the top of the canopy crown (Altimir et al., 2006). The conductance is derived from the vapor pressure deficit at leaf temperature assuming that the resistance due to the leaf boundary layer is negligible due to

ventilation of the air in the shoot chambers. The leaf temperature is calculated following a leaf energy balance model that incorporated heating by incoming shortwave radiation, cooling by transpiration and convection, and thermal radiation balance. Conductances measured under humid conditions (relative humidity (RH) > 80 %) were rejected due to the underestimation of transpiration at higher RH levels. The stomatal conductance to COS ($g_{sCOS}$) is derived based on the relationship between COS and H$_2$O conductance: $g_{sCOS} = g_{sw}/R_{wCOS}$ (Seibt et al., 2010) where $R_{w-COS}$ is the ratio of H$_2$O

and COS diffusivities and is derived by Seibt et al. (2010) to be $2.0 \pm 0.2$.

### 2.4.3 Meteorological data

Meteorological data such as the friction velocity ($u_*$), air temperature ($T_{air}$), relative humidity (RH), soil water content (SWC) and wind direction were available through the SmartSMEAR database which contains continuous data records from the SMEAR sites (available at http://avaa.tdata.fi). The vapour-pressure deficit (VPD) was calculated from RH and $T_{air}$.

## 3 Flux derivations

### 3.1 The EC-based method

### 3.1.1 Eddy covariance fluxes

The EC technique is based on turbulence measurements above the canopy and fluxes are derived from the covariance between a scalar (in this case COS or CO$_2$) and the vertical wind speed (e.g. Aubinet et al., 2012; Mammarella et al., 2007).

The fluxes derived through this method represent the net exchange of gases between the canopy layer and the air above. The EC technique requires turbulent conditions, otherwise gases that accumulate or get depleted due to sources and/or sinks within the canopy do not reach the sensors above the canopy. As soon as turbulence is enhanced in the early morning, these gases are released to levels above the canopy and are only then being captured by the EC system. This so called storage change within the canopy can be significant and should be added to the turbulence flux to account for the delayed capture of



fluxes by the EC system (Aubinet et al., 2012). In this study we refer to the storage-corrected COS and $CO_2$ EC flux as $F_{COS-EC}$ and $NEE_{EC}$, respectively. The calculation of storage fluxes is discussed in the next section. In this study the EC fluxes were calculated using the EddyUH software package developed at the University of Helsinki (Mammarella et al., 2016). In short, the high-frequency EC data were despiked according to standard approach (Vickers and Mahrt, 1997). The

spectroscopic correction due to $H_2O$ impact on the absorption line shape was accounted for along with the dilution correction in the QCLS acquisition software. A 2D rotation of sonic anemometer wind components was performed, and 30 min covariances between the scalars and vertical wind velocity were calculated using linear detrending method. Short-term drift in the QCLS high-frequency concentration data was negligible and there was no need to apply more sophisticated approach for detrending the data, e.g. high pass recursive filters (Mammarella et al., 2010). The time lag between the concentration

and wind measurements induced by the sampling line was determined by maximizing the covariance. Due to better signal-to-noise ratio, the lag for COS was determined by maximizing the covariance for QCLS $CO_2$, and the same lag was assigned to COS. Finally, spectral correction was done according to Mammarella et al. (2009). Total random uncertainty of the fluxes (Rannik et al., 2016) was calculated according to the method implemented in EddyUH, the method proposed by Finkelstein and Sims (2001). The uncertainties of $NEE_{EC}$ and $F_{COS-EC}$ are estimated from the standard deviation of data points per night,

where night is defined as the time when the sun elevation angle is below -3°. A general observation that is seen with EC measurements is that nighttime $NEE_{EC}$ decreases with lower $u_*$, whereas respiration is not expected to depend on atmospheric turbulence. For this reason we filtered out (storage-corrected) fluxes with $u_*$ values below a threshold of 0.3 m s$^{-1}$ (Mammarella et al., 2007). A difference between COS and $CO_2$ fluxes is, however, that the uptake of COS by leaves is concentration dependent (Berry et al., 2013) and the leaf boundary layer may get depleted in COS under low turbulence

conditions, slowing uptake rates. It is unknown to what extent this affects COS fluxes in practice, but it has to be kept in mind that the $u_*$ filtering may be an overstated filtering to COS fluxes. To determine the fraction that nighttime COS fluxes contribute to total daily COS uptake we gapfilled COS fluxes with a rectangular hyperbola light response function that is based on the measured data. Missing COS data under dark conditions were filled based on the average nighttime flux obtained from this study.


$CO_2$ and $H_2O$ ecosystem fluxes from the QCLS were compared with those from the nearby tall tower. During nighttime, the QCLS $CO_2$ flux is a factor 0.73 smaller than the tall-tower fluxes at the same height and the underestimation has been observed with another EC-system at the small tower as well. Kolari et al. (2009) found that the tall tower $NEE_{EC}$ agrees well with upscaled soil and branch chamber measurements. As we rely on the accuracy of $NEE_{EC}$ in the radon-tracer method

(Section 3.2) we use $NEE_{EC}$ from the tall tower instead of the QCLS at the smaller tower throughout the manuscript. The underestimation is not the same for all gases, e.g. the evapotranspiration flux is only a factor 0.97 smaller. It is therefore unknown by how much the $F_{COS-EC}$ flux is affected by the general underestimation at the small tower.



### 3.1.2 Storage fluxes

Storage fluxes ($F_{stor}$) are defined as the integral of concentration changes over height up to the height of the EC measurements ($h_{EC}$):

$$F_{stor} = \frac{P}{RT_{air}} \int_0^{h_{EC}} \frac{dC(z)}{dt} dz$$

with $P$ the atmospheric pressure, $R$ the molar gas constant and $C(z)$ the COS or $CO_2$ concentrations (ppt for COS or ppm for $CO_2$) along a profile (Aubinet et al., 2001; Papale et al., 2006). The integral was determined from hourly measured profile concentrations at 0.5, 4, 14, and 23 m in two ways: (1) by integrating an exponential fit through the data, and (2) using trapezoidal areas (Winderlich et al., 2014). The concentration at ground level that is used for the second calculation method is estimated by extrapolating the gradient between 0.5 and 4 m to the ground level. A third calculation was done assuming a constant profile from the EC measurement height (23 m) to the ground level, to test the bias in storage fluxes when no profile measurements are available. The results of the different calculation methods will be discussed in Section 4.1. To reduce the error due to the random noise of COS concentration measurements, a running average over a 5 hour window was applied to the COS concentration data before the storage fluxes were calculated.

### 3.2 The radon-tracer method

$^{222}$Rn is a natural radioactive gas that is formed by the decay of $^{226}$Radium, which is uniformly distributed in soils (Van der Laan et al., 2009). Once in the atmosphere, $^{222}$Rn is affected by radioactive decay and atmospheric mixing. As the exhalation rate of $^{222}$Rn by the soil ($F_{Rn}$) is considered constant and uniformly distributed, and $^{222}$Rn is mixed through the atmosphere in the same way as other trace gases, the surface fluxes of these trace gases ($F_C$) can be determined from the concentration change of these gases over time ($\Delta C$) relative to that of $^{222}$Rn ($\Delta^{222}$Rn) (Schmidt et al., 1996; Van der Laan et al., 2009; Belviso et al., 2013):

$$F_C = F_{Rn} \frac{\Delta C}{\Delta^{222}Rn}$$

$^{222}$Rn generally builds up in the boundary layer when it gets shallower during the night. Fig 1. shows an example of one night during the measurement campaign where $^{222}$Rn concentrations increase in the evening and reach a maximum in the night, while at the same time $CO_2$ increases and COS decreases. This nighttime buildup of gases and the constant surface flux of $^{222}$Rn make the radon-tracer method appropriate to derive nighttime fluxes of trace gases. Requirements for this method are that the $^{222}$Rn concentrations are corrected for radioactive decay, that $F_{Rn}$ is known, and that a high correlation exists between the trace gas and $^{222}$Rn concentrations. Moreover, when the spatial distribution of sources and sinks of a trace gas are similar to the source of $^{222}$Rn at the ground, a high correlation between the trace gas and $^{222}$Rn can be expected. Therefore, the correlation between COS and $^{222}$Rn concentrations may give insight into the distribution of sinks of COS within the ecosystem.



One of the main uncertainties of the radon tracer method is the magnitude of $F_{Rn}$. In Szegvary et al. (2007), $F_{Rn}$ was measured at a site 46 km away from the SMEAR II site, which resulted in $F_{Rn}$ = 15.3 mBq m$^{-2}$ s$^{-1}$. Model studies have estimated $F_{Rn}$ in Europe from 4.0 to 12.4 mBq m$^{-2}$ s$^{-1}$, (summarized in Table S1 in supplementary material), leading to an overall average of 9.6 ± 4.1 mBq m$^{-2}$ s$^{-1}$. The exhalation rates depend on the uranium content and soil properties that affect

diffusive transport such as the soil texture and soil moisture (Karstens et al., 2015). The $F_{Rn}$ values of 4.0 and 11.4 mBq m$^{-2}$ s$^{-1}$ that were modelled by Karstens et al., 2015 for two different soil moisture maps indicate that the uncertainty of $F_{Rn}$ is in large part caused by different soil moisture.

As the uncertainty of the COS and $CO_2$ ecosystem fluxes derived through the radon-tracer method ($F_{COS-Rn}$ and $NEE_{Rn}$

respectively) is in large part determined by the uncertainty of $F_{Rn}$, it is key to further limit the $F_{Rn}$ range between 4.0 and 15.3 mBq m$^{-2}$ s$^{-1}$ in Table S1. For that reason we inverted the radon-tracer method to derive $F_{Rn}$ from $CO_2$ and $^{222}$Rn concentrations with a known ecosystem $CO_2$ flux ($NEE_{EC}$), instead of a known $F_{Rn}$ to derive NEE, which is normally used in the radon-tracer method (Van der Laan et al., 2016). The advantage of this method is that $F_{Rn}$ is obtained from actual measurements at the site, and we will therefore use this $F_{Rn}$ to determine $F_{COS-Rn}$. The $F_{Rn}$ that we derived in this way is 5.2

mBq m$^{-2}$ s$^{-1}$ with a standard deviation of 2.7 mBq m$^{-2}$ s$^{-1}$ and a standard error of 0.47 mBq m$^{-2}$ s$^{-1}$. This value of $F_{Rn}$ is within the range listed in Table S1, but is lower than the average of 9.6 mBq m$^{-2}$ s$^{-1}$. We will discuss in Section 5.2 what the effect of canopy layer mixing can be on the derivation of $F_{Rn}$ and COS fluxes. Temporal variation of $F_{Rn}$ can be expected due to the changes in SWC that affects the soil permeability; however, no temporal change or correlation with SWC was found ($R^2$ = $2 \cdot 10^{-5}$) throughout the season.

In Hyytiälä, $^{222}$Rn measurements were made at 8 m, and COS and $CO_2$ concentrations from the same height need to be used to derive their surface fluxes. We derived concentrations at 8 m from an exponential fit through the profile concentrations at 0.5, 4, 14 and 23 m. A linear fit between 4 and 14 m was used in cases where the algorithm for the exponential fit did not converge. The factor $\Delta C / \Delta^{222}$Rn is determined from a linear regression of concentrations of COS or $CO_2$ against $^{222}$Rn. Data

that are used for the linear regression fall between the minimum $^{222}$Rn concentration in the late afternoon and maximum $^{222}$Rn concentration in the night (see Fig. 1 for an example). Per night, a minimum of 5 data points need to be available and $R^2$ between $^{222}$Rn and $CO_2$ and COS should be at least 0.5 (for $CO_2$) and 0.3 (for COS). Uncertainties of $NEE_{Rn}$ and $F_{COS-Rn}$ are determined from the linear regression as the standard error of the slope.

### 3.3 Soil fluxes

Soil fluxes ($F_{soil}$) were calculated from least square fits of the concentrations during chamber closure and by considering mass balance equations within the chamber (Sun et al., 2017). At the start of the campaign we did blank tests by placing FEP foil over the soil and calculated fluxes through the standard measurement procedure. Soil fluxes were corrected for blank





chamber effects of 0.66 ± 0.48 pmol m$^{-2}$ s$^{-1}$ for COS, blanks for $CO_2$ were negligible (-0.05 ± 0.15 μmol m$^{-2}$ s$^{-1}$). Further details about the soil flux measurements can be found in Sun et al. (2017).

## 4. Results

### 4.1. COS and $CO_2$ storage fluxes

The storage fluxes of COS (Fig. 2) are slightly negative during nighttime with an average nighttime value of -0.7 pmol m$^{-2}$ s$^{-1}$. Early in the morning when turbulence is enhanced, the storage fluxes become positive and have an average maximum of 1.9 pmol m$^{-2}$ s$^{-1}$ at 09:00. The storage fluxes of $CO_2$ follow a similar pattern but have the opposite sign. Storage fluxes of COS calculated from trapezoidal areas are on average 25 % larger than when an exponential fit through the profile is integrated. When the concentration profile is assumed to be constant from the EC measurement height to the ground level, the storage flux is on average 7 % smaller compared to a profile with an exponential fit. These differences are small compared to the size of the ecosystem fluxes. Neglecting storage fluxes would not influence the long-term budget of COS and $CO_2$, as it only corrects for the delay in release of accumulated gases from within the canopy (Aubinet et al., 2012); however, it does affect the diurnal variability of fluxes, and any attempt at flux partitioning, particularly if storage fluxes are large. In this dataset, storage fluxes of both COS and $CO_2$ are small compared to the EC flux, i.e. storage fluxes are on average 5 % of $F_{COS-EC}$ and 7 % of $NEE_{EC}$.

### 4.2 COS and $CO_2$ nighttime fluxes through the radon-tracer and EC-based method

The linear correlation between the concentrations of $^{222}$Rn and the scalar (COS or $CO_2$) is key in interpreting the fluxes derived from the radon-tracer method. Fig. 3 shows the distribution of $R^2$ values for the correlation between $^{222}$Rn and COS or $CO_2$. The correlation between $^{222}$Rn and $CO_2$ peaks at $R^2$ values in the range 0.9-1.0 and has a median value of 0.79. The $R^2$ for COS is generally lower with a median of 0.59. The lower $R^2$ values for COS can partly be explained by the lower precision of COS measurements compared to those of $CO_2$. However, the average $R^2$ only slightly increases to 0.64 when the noise of COS is diminished by taking a running average of a 5 hour window over the COS measurements. This indicates that the lower precision of COS is not the main aspect influencing the correlation with $^{222}$Rn. Another aspect that influences the correlation with $^{222}$Rn is the similarity in vertical distribution of sources and sinks between the scalar and $^{222}$Rn, which will be further discussed in Section 5.1.

The radon-based nighttime fluxes of COS and $CO_2$ are compared with the EC-based fluxes in Fig. 4. $F_{COS-Rn}$ ($NEE_{Rn}$) was determined for 79 (87) out of 128 nights during the campaign that passed the criteria of a minimum $R^2$ and a minimum number of available data. Nighttime fluxes derived with the EC method were determined for 56 nights following removal of 43% of the data due to $u_*$ filtering. $F_{Rn}$ was derived from $^{222}$Rn concentrations in relation to $NEE_{EC}$ and $CO_2$ concentrations in order to limit the uncertainty of $F_{Rn}$ on $F_{COS-Rn}$. This means that the average $NEE_{EC}$ and $NEE_{Rn}$ values are close (3.30 ± 0.62




and 3.31 ± 0.48 µmol m$^{-2}$s$^{-1}$ respectively) as they are not independent from each other. Both NEE$_{EC}$ and NEE$_{Rn}$ show a decreasing trend from summer towards autumn. However, the R$^2$ value between NEE$_{EC}$ and NEE$_{Rn}$ is only 0.06, which is likely due to the low signal-to-noise ratio of both flux techniques.

Both the EC-based and radon-tracer methods show negative nighttime COS fluxes with an average of -7.9 ± 3.8 pmol m$^{-2}$s$^{-1}$ (F$_{COS-EC}$) and -8.1 ± 1.5 pmol m$^{-2}$s$^{-1}$ (F$_{COS-Rn}$). In comparison, nighttime soil fluxes of COS are on average -2.7 pmol m$^{-2}$ s$^{-1}$ (Fig. 2) and soil fluxes do not show a clear diurnal or seasonal cycle. An overview of the soil fluxes is presented in Sun et al. (2017). Similar to NEE, a decreasing trend is visible in both F$_{COS-Rn}$ and F$_{COS-EC}$ with an average of -10.9 pmol m$^{-2}$ s$^{-1}$ in July and -7.1 pmol m$^{-2}$ s$^{-1}$ in October as obtained from F$_{COS-Rn}$. The nighttime uptake is 38 % of the average daytime fluxes
(defined as when sun elevation is above 20°) and 21 % of the total daily COS uptake (obtained from gapfilled data). When the soil flux is subtracted from the ecosystem flux, the nighttime uptake is 17 % of the total daily uptake.

### 4.3 F$_{COS}$ correlation with g$_{sCOS}$, VPD, T$_{air}$ and u$_*$

Fig. 5 shows F$_{COS}$ against nighttime averaged g$_{sCOS}$, VPD, T$_{air}$ and u$_*$ with their respective uncertainties. Soil fluxes did not show a seasonal or daily cycle (Sun et al., 2017) and are therefore not subtracted from the ecosystem-scale fluxes, as this
would only add noise to the fluxes. The nights shown in Fig. 5 only cover summer nights between June 28 and August 25, 2015, as g$_{sCOS}$ data did not pass the RH filter criteria after this period due to higher RH. The month August was characterized by a dry period with SWC decreasing from about 20 % down to 7 %, the average nighttime temperature increased and RH decreased. Over the same time period, nighttime g$_{sCOS}$ decreased from 0.02 mol m$^{-2}$ s$^{-1}$ to 0.006 mol m$^{-2}$ s$^{-1}$ (see Fig. S1 in supplementary material for an overview of the meteorological conditions).

Weak correlations are found between F$_{COS-Rn}$ and g$_{sCOS}$ (R$^2$ = 0.43), T$_{air}$ (R$^2$ = 0.43) and VPD (R$^2$ = 0.24) where fluxes decrease under lower g$_{sCOS}$ and higher VPD and T$_{air}$. No correlation was found with u$_*$ (R$^2$ = 0). The same comparison was made for F$_{COS-EC}$ (Fig. S2 in supplementary material), which gave correlations R$^2$ = 0.36 (g$_{sCOS}$), 0.30 (T$_{air}$), 0.56 (VPD) and 0.50 (u$_*$) and showed that also F$_{COS-EC}$ decreased under lower g$_{sCOS}$, and higher VPD and T$_{air}$. However, these correlations
were only found when no u$_*$ filter was applied, as only a few data points remained after the u$_*$ filtering.

g$_{sCOS}$ was on average 0.016 mol m$^{-2}$ s$^{-1}$ during nighttime and 0.117 mol m$^{-2}$ s$^{-1}$ during daytime. The average nighttime g$_{sCOS}$ showed a correlation with the average nighttime VPD (R$^2$ = 0.54, not shown) and g$_{sCOS}$ was negatively correlated with T$_{air}$ (R$^2$ = 0.60; not shown).





## 5. Discussion

### 5.1 Vertical distribution of sinks and sources of COS and $CO_2$ compared to that of $^{222}$Rn

The benefit of stable conditions within the canopy layer is that the correlation of COS or $CO_2$ with $^{222}$Rn can shed light on the spatial distribution of sources and sinks of these gases in comparison to the only source of $^{222}$Rn, which is the soil. When

the source or sink of COS or $CO_2$ is focused at the ground level, a high correlation between $^{222}$Rn and these gases can be expected. The fact that $CO_2$ shows a high correlation with $^{222}$Rn indicates that the main source of $CO_2$ is located near the surface, which is confirmed by the magnitude of nighttime soil chamber measurements relative to branch chamber measurements in Kolari et al. (2009), who found that respiration of the tree foliage was 1.5 - 2 µmol m$^{-2}$ s$^{-1}$ during summer nights and soil respiration was 5 - 6 µmol m$^{-2}$ s$^{-1}$. In contrast, we find that the correlation between $^{222}$Rn and COS is lower,

which suggests that the main sink of COS is not near the surface, but rather at higher levels in the canopy layer. This is also supported by the soil chamber measurements, which suggest that the soil contributes to 33 % of the total nighttime COS uptake.

### 5.2 The effect of canopy layer mixing on flux derivations

When the canopy air is fully mixed, the flux obtained through the radon-tracer method represents the net exchange flux in

that canopy layer, regardless of the potential difference in the spatial distribution of the tracer fluxes, e.g. $CO_2$ and $^{222}$Rn. In this study, however, the $^{222}$Rn concentrations are measured within the canopy layer at 8 m and decoupling of canopy layers may exist (Alekseychik et al., 2013). Fluxes derived from concentrations within the canopy may therefore not represent the exchange of these gases in the whole canopy. To discuss the effect of decoupling on radon-flux calculations we have to distinguish between two decoupling situations; (1) when the 8 m air is decoupled from the air close to the ground, and (2)

when the 8 m air is decoupled from the canopy layer above:

1.   When the 8 m canopy layer is decoupled from the air close to the ground, the different flux distribution of $CO_2$ and $^{222}$Rn can become apparent. In the case of decoupling, the respiration of the tree foliage would influence the 8 m concentration, while the $CO_2$ respiration and radon flux at the surface do not influence the air at 8 m. The 8 m concentration is then not representative for the canopy layer $CO_2$ flux and would lead to a lower $F_{Rn.}$ This would explain

why the $F_{Rn}$ that we find (5.2 mBq m$^{-2}$ s$^{-1}$) is lower than the average $F_{Rn}$ reported in other literature (9.6 ± 4.1 mBq m$^{-2}$ s$^{-1}$). At the same time, when COS fluxes do not entirely take place at the surface but within the canopy, this would lead to a higher $F_{COS-Rn}$.

2.   When the 8 m layer is decoupled from the canopy layer above, the air that is depleted in COS due to the sinks within the canopy may not reach the lower canopy layers on which $F_{COS-Rn}$ is based and leads to an underestimation of $F_{COS-Rn}$.

Furthermore, the decoupled layer at the surface is more susceptible to horizontal advection which may affect the concentration profile as well.





Alekseychik et al. (2013) identified decoupling of different canopy levels at the Hyytiälä site based on changing wind directions at different heights. They observed a decrease in $NEE_{EC}$ under decoupled circumstances, which occurred in at least 18.6 % of all nighttime periods. We did not observe a correlation with $F_{COS-Rn}$ and the difference in wind direction between 16.8 and 8.4 m. However, a limitation is that we can only compare nighttime averages, whereas decoupling does not have to last throughout the whole night and can also exist during only a fraction of a night. Furthermore, we do not have wind direction data at other heights within the canopy to be able to determine if the decoupling takes place below or above 8 m.

### 5.3 Sensitivity of $F_{COS-EC}$ to $u_*$.

It is well accepted that $NEE_{EC}$ underestimates the true NEE under low $u_*$, as nighttime NEE (respiration only) is not expected to depend on atmospheric turbulence. By applying a $u_*$ filter to COS fluxes, we assume the same independence of COS uptake to atmospheric turbulence. However, a negative correlation between $F_{COS}$ and $u_*$ can be expected when the leaf boundary layer gets depleted in COS under low turbulence conditions and the uptake of COS gets limited by the COS gradient at the leaf boundary layer. If this is the case, that means that by applying the $u_*$ filtering to $F_{COS-EC}$ we bias to higher $F_{COS-EC}$ data. The dependence of $F_{COS-EC}$ to $u_*$ ($R^2 = 0.50$) is not observed for $F_{COS-Rn}$ ($R^2 = 0.00$), which suggests that $F_{COS}$ is not limited by the COS gradient at the leaf boundary layer and that the lower $F_{COS-EC}$ under low $u_*$ is a real measurement artifact. Still, the fact that we find relations of $g_{sCOS}$ and $T_{air}$ with $F_{COS-EC}$ (when the $u_*$ filter is not applied) that are similar to relations with $F_{COS-Rn}$, may be an indication that the $u_*$ filtering is an overstated filtering. Unfortunately we cannot determine if the effect from $u_*$ on $F_{COS-EC}$ is due to limitations of the EC method or due to actual reduced COS uptake by the leaves under low $u_*$.

### 5.4 Stomatal control of nighttime $F_{COS}$

A correlation between nighttime $F_{COS}$ and $g_{sCOS}$ was expected due to stomatal diffusion and the light-independence of the CA enzyme. A correlation of $g_{sCOS}$ with $F_{COS}$ was indeed observed for both the radon-tracer and EC method, although the latter was only found when no $u_*$ filtering was applied to the data, as only a few data points remained when the $u_*$ filtering was included. The decrease in $F_{COS}$ when $g_{sCOS}$ decreases and VPD increases is likely related to the dry and warm period in August to which plants respond by closing their stomata to prevent excessive water loss. This would also explain why $F_{COS}$ is lower under high $T_{air}$. In general we do not find strong correlations between the COS flux and the nighttime environmental parameters, which can be explained by the low signal-to-noise ratio of the flux measurements and the fact that $F_{COS-Rn}$ may not represent the full canopy layer due to decoupling (see Section 5.2). Moreover, we compare ecosystem fluxes with leaf-level $g_{sCOS}$ within enclosed chambers, which may not represent the full canopy dynamics. Nevertheless, the fact that both the radon-tracer and the EC methods confirm that the COS uptake decreases with decreasing $g_{sCOS}$ indicates that the nighttime uptake of COS is indeed driven by vegetation. Moreover, soil fluxes were found to be -2.7 pmol m$^{-2}$ s$^{-1}$ on average. With the total nighttime COS uptake being –7.9 to -8.1 pmol m$^{-2}$ s$^{-1}$, soil fluxes contribute to only 33 % of the nighttime COS uptake.



Besides uptake of COS by the soil and leaf stomatal diffusion there is no other process to our knowledge that would lead to uptake of COS in the ecosystem. This leads to the conclusion that the nighttime COS uptake is predominantly driven by vegetative uptake and supports the use of COS to estimate $g_{sCOS}$ (Wehr et al., 2017). Assuming that the soil is the only sink besides the vegetation, we can say that the nighttime vegetative uptake contributes to 17 % of the total daily COS uptake.

Moreover, this study has confirmed that nighttime stomatal conductance exists at the Hyytiälä site.

**5.5 Effect of nighttime COS fluxes on GPP derivation**

The measurements in this study showed that, unlike the uptake of $CO_2$, the COS uptake continues during the night, which agrees with the light-independence of the CA enzyme. We showed that the nighttime plant COS fluxes cover 17 % of the total daily COS plant uptake, which indicates that nighttime COS uptake is a significant sink in the total COS budget.

Including this nighttime sink is essential in regional COS models and will affect COS-based GPP model simulations as well. The relationships that we found between $F_{COS}$, $g_{sCOS}$, VPD and $T_{air}$ will aid in implementing nighttime $F_{COS}$ in models. Furthermore, the light-independence of COS uptake should be taken into account when COS is being used as tracer for GPP. Besides restricting COS as GPP-tracer to light conditions, the leaf relative uptake ratio (LRU), which is the normalized ratio between COS and $CO_2$ fluxes, can be expected to increase when GPP becomes zero around sunrise and sunset while at the

same time COS is continuously being taken up by vegetation. So far, only Stimler et al. (2011) showed the light-dependence of LRU from leaf-scale measurements and Maseyk et al. (2014) observed a light-dependence of LRU which was derived from soil and ecosystem fluxes. Other studies have focused on LRU values under high light conditions (e.g. Sandoval-Soto et al., 2005; Berkelhammer et al., 2014). More leaf-level COS flux measurements should be made to accurately parameterize the light-dependence of LRU in the field.

**6. Conclusion**

In this study we quantified nighttime COS fluxes in a boreal forest using both the EC and the radon-tracer methods, and found that nighttime $F_{COS}$ between June and November 2015 was on average -7.9 ± 3.8 pmol m$^{-2}$ s$^{-1}$ and -8.1 ± 1.5 pmol m$^{-2}$ s$^{-1}$ according to the two different methods, respectively. A high correlation between $CO_2$ and $^{222}$Rn indicates that the sources of these gases have a similar spatial distribution, namely at the soil. A lower correlation of $^{222}$Rn with COS suggests that the

main sink of COS is not located at the surface, but rather at higher levels in the canopy. This is supported by soil chamber measurements, which show that the soil flux is on average -2.7 pmol m$^{-2}$ s$^{-1}$ and only contributes to 33 % of the total nighttime COS uptake.

Our estimates for nighttime $F_{COS}$ are 38 % of the size of daytime average $NEE_{EC}$ fluxes. Based on the EC method, the

nighttime COS uptake is 21 % of the total daily COS uptake and is mostly driven by aboveground vegetation. Furthermore, we investigated the relation of the nighttime COS fluxes with stomatal conductance ($g_{sCOS}$) and environmental parameters.





Measurements of both $F_{COS-Rn}$ and $F_{COS-EC}$ pointed to a decrease of COS uptake with decreasing $g_{sCOS}$ and increasing VPD and $T_{air}$, which is likely related to a dry and warm period in August to which plants responded by closing their stomata to prevent excessive water loss. Our results suggest that the nighttime uptake of COS is mainly driven by the tree foliage and the relationships that we find between $F_{COS}$, $g_{sCOS}$, VPD and $T_{air}$ will aid in implementing nighttime COS uptake in models.

Both the EC and the radon-tracer methods indicate that the nighttime sink of COS plays an important role in the total COS budget in a boreal forest and needs to be taken into account when using COS as a tracer for GPP estimates.

**Author contributions**

U.S., K.M., H.C., T.V. and L.M.J.K designed the research. L.M.J.K., K.M., J.A. conducted the field work, W.S., I.M., P.K., A.F, R.V and G.V provided data. L.M.J.K performed data analysis. L.M.J.K. and H.C. wrote the paper with contributions

from all co-authors.

**Competing interests**

The authors declare that they have no conflict of interest.

**Acknowledgement**

We greatly appreciate the maintenance and help of the technical staff at SMEAR II in Hyytiälä, in particular Helmi Keskinen

and Janne Levula. We also thank Bert Kers and Marcel de Vries for their help during preparations of the campaign at the University of Groningen. We would like to thank Ute Karstens and Navin Manohar for making $F_{Rn}$ simulations and corresponding data available. The research leading to these results has received funding from the European Community's Seventh Framework Programme (FP7/2007-2013) in the InGOS project (n° 284274), the NOAA contract NA13OAR4310082, the Academy of Finland Centre of Excellence (n° 118780), Academy Professor projects (n°

284701 and 282842), ICOS-Finland (n° 281255) and CARB-ARC (n° 286190).

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



**Figures**

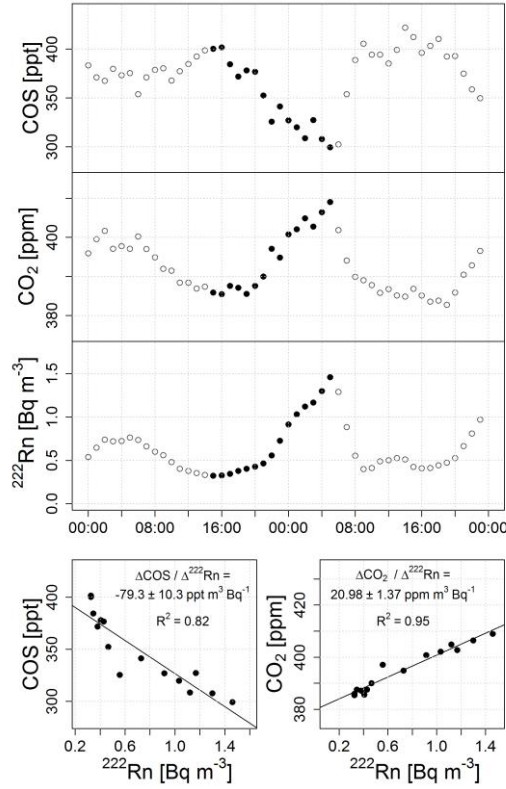

Figure 1: COS, CO2 and 222Rn concentrations during 12-13 July 2015 where the data between the minimum and maximum
222Rn concentration are used to derive nighttime fluxes of COS and CO2 (black, filled). The bottom figures show the linear
regression between 222Rn and COS (left) and CO2 concentrations (right) on which FCOS-Rn and NEERn are based.





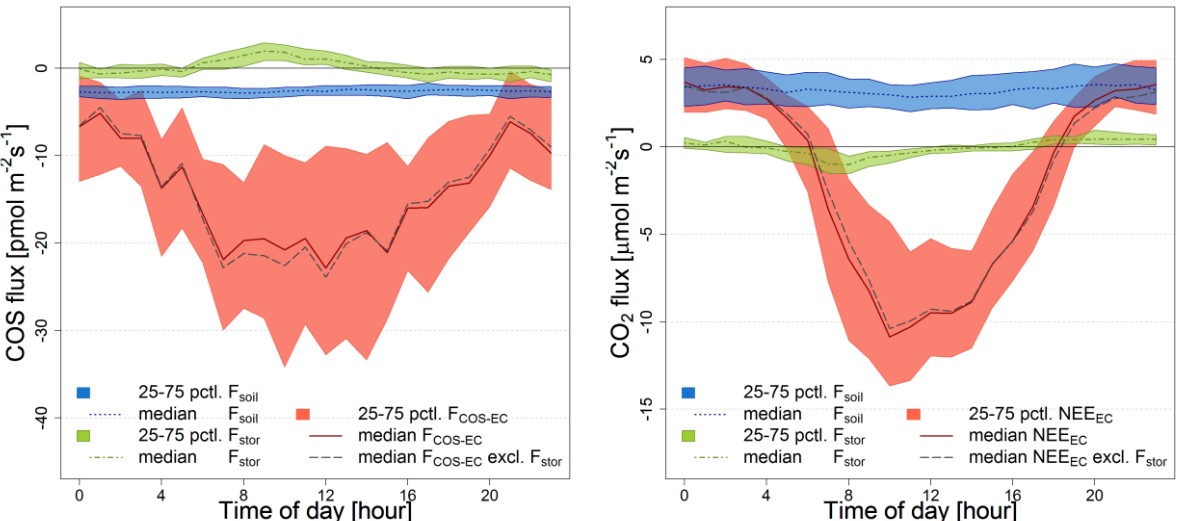

Figure 2: Storage fluxes $F_{stor}$ (green), ecosystem fluxes $NEE_{EC}$ and $F_{COS-EC}$ (red) and soil fluxes $F_{soil}$ (blue) of COS (left) and $CO_2$ (right). Thick lines indicate the median values of the data over the whole measurement period, and the shaded areas specify the $25^{th}$-$75^{th}$ percentiles. The median values of $NEE_{EC}$ and $F_{COS-EC}$ without storage correction are shown in gray. The ecosystem fluxes are filtered for low $u_*$ values with a threshold of 0.3 m s$^{-1}$.





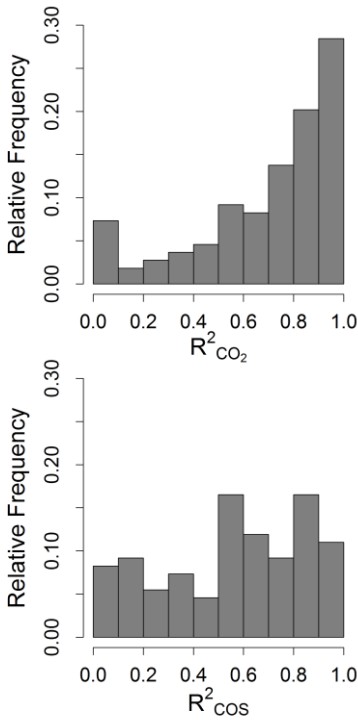

Figure 3: Relative frequency of $R^2$ values of the correlation between concentrations of $^{222}$Rn and $CO_2$ (top) and COS (bottom).




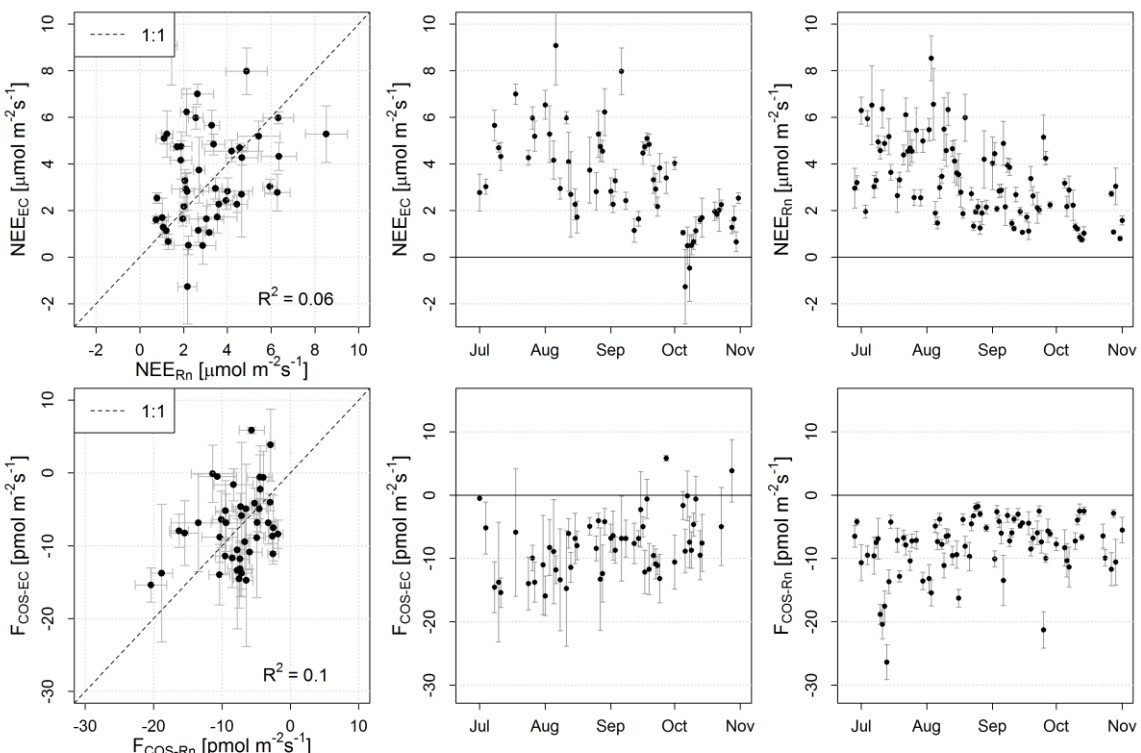

Figure 4: Left: comparison of EC- and radon-based fluxes for average nighttime $CO_2$ (top) and COS (bottom) fluxes. Middle and right: time series of EC based fluxes (middle) and radon-based fluxes (right). The uncertainty bars of the EC and radon-based fluxes are not directly comparable due to the different ways of determining these uncertainties.





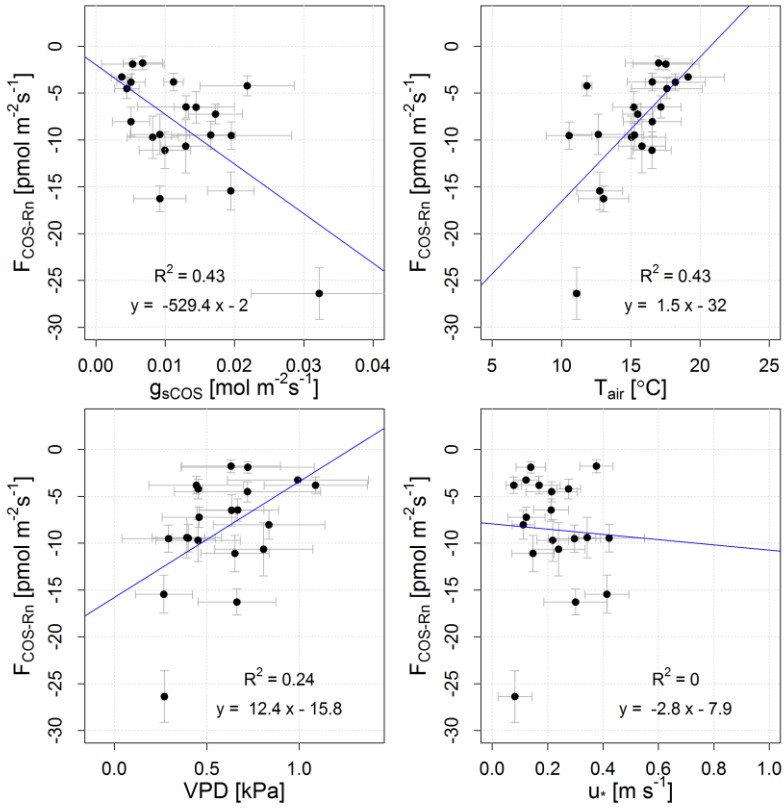

Figure 5: Correlations of $F_{COS-Rn}$ with $g_{sCOS}$, $T_{air}$, VPD, and $u_*$. All data (except $F_{COS-Rn}$) are averages over individual nights (with nighttime defined as sun elevation below -3°). Data in this plot largely represent a period in August 2015 with dry conditions (i.e. decreasing SWC, and increasing $T_{air}$ and VPD).

