# Peer review of "Canopy uptake dominates nighttime carbonyl sulfide fluxes in a boreal forest"

_Atmospheric Chemistry and Physics, 2017_

## Referee Comment (RC1) · Anonymous Referee #1 · 15 Jun 2017

In this study, the authors profiled COS and $CO_2$ concentrations at 5 heights for 5 months in 2015 for evaluating storage fluxes and understanding the processes of gas exchange, concomitantly with eddy covariance and radon measurements over and in the canopy for assessing the vertical fluxes. Special attention is paid to the nighttime uptake of COS and to the apportionment of this sink within the ecosystem. I share the conclusions of this paper which is well written and deserves to be published, but more detailed information is required in an area of major importance to the study, i.e. the role of plants in the nighttime uptake of COS which, in this manuscript, is only assessed indirectly (i.e., Plant flux = Total flux – Soil flux) because the authors make very little use of their short-term COS profile measurements. If trees are a larger sink of COS than soils during the night, there should be some sign of COS drawdown at canopy level

especially if the 8 m canopy layer is decoupled from the air close to the ground as discussed in chapter 5.2. I look forward seeing new plots showing isolines of the average COS concentration distributions within the canopy as a function of height (including the 125 m reference level) and time of day for the summer and autumn months (see my comment about Fig. 2 below). I hope that this analysis will not end up showing that there are no clear vertical changes in COS between .5 and 125 m height during the night.

Methods

The authors used a multi-position Valco valve to switch frequently (10 times per hour) between the sample tubing from the different profile heights. It would be useful to know the flow rates through the sampling lines (are they flushed permanently or not?) and through the QCLS sampling cell which internal volume could be reminded. Did you use data from the last xx seconds of each cycle or the 3 min records? Did you notice memory effects from previous samples? I would highly recommend the authors to show in a new figure a typical 1h cycle recorded in the late night (stable atmospheric conditions favoring COS and CO2 stratification) and in the afternoon (vertical mixing, no vertical gradient).

Figures

Fig. 1 provides a nice illustration of the radon-tracer method but the times of sunrise and sunset are missing. I guess that a significant portion of daytime Rn measurements is used to calculate the linear regressions shown in the lower panels from which the nighttime fluxes of COS and CO2 are derived. This appears inconsistent to me. 8 m data extrapolated from other levels using an exponential fit isn't it? I also suggest adding the diurnal variations of hourly values of storage fluxes and friction velocity during 12-13 July 2015.

Fig. 2 shows the mean diurnal variations in fluxes based on all available data with friction velocities > to 0.3 m/s (a quite high threshold to separate stable from turbulent

atmospheric conditions). Such a presentation is inadequate because daylight duration from early July to late October exhibits large variations at 61°N as stated page 3 line 15. I don't think it is necessary to generate monthly averages of hourly fluxes, averaged values for summer and autumn months would be adequate.

Abstract

Page 1 line 18: the total nighttime COS fluxes over the whole measurement period were. . .

Page 1 line 21: . . .suggesting that the main sink of COS is not located at the ground. May be the new analysis of vertical profiles will demonstrate that the main sink of COS is not located at the ground.

---

## Referee Comment (RC2) · Anonymous Referee #2 · 29 Jun 2017

In the article "Canopy uptake dominates nighttime carbonyl sulfide fluxes in a boreal forest" Kooijmans and co-authors present a season of nighttime fluxes of COS and CO2 derived at a height of 8m in a boreal forest with a dominant canopy height of 17 m. Fluxes are derived by eddy covariance, but recognizing the limitations of eddy covariance under placid nighttime conditions, the authors derive fluxes by gradient-flux similarity methods to Radon 222, which is emitted at consistent rates from soil. The authors find evidence for significant nocturnal uptake of COS by the canopy, suggesting a greater role of vegetation than soils in atmospheric COS uptake both during the day and night at this site. The measurement methods and analysis are thorough, and the results provide much needed data to the field including independent measurements of stomatal conductance for comparison with COS fluxes. This is a valuable contribution

to understanding the behavior of COS in ecosystems for more precise application as a carbon cycle tracer. General and specific comments follow.

General comments:

The manuscript discusses the possibility that under still conditions, when eddy covariance techniques are not applied due to low u*, COS may be depleted at the leaf surface and slow uptake rates. This is discussed in relation to the suitability of u* filtering. However, a similar phenomenon could occur at the soil-atmosphere interface under still conditions where COS uptake rates are limited by COS availability in depleted layers low in the profile. Under those conditions, emissions of $CO_2$ and $222Rn$ would however not be limited given that they are production reactions. How would concentration-depletion at the soil-atmosphere interface affect interpretation of the data in this paper (for example the interpretation of Figure 3)?

It would be useful to discuss the uncertainty in scaling up soil flux measurements from the chamber measurements. How much variation was there between chambers? Given the large difference in footprint between tower-based and chamber measurements, how could spatial heterogeneity affect your estimations of the role of nocturnal canopy uptake of COS?

No significant trend of F_Rn derived from NEE was reported with SWC, but was there a trend over the season? I would find a time series of F_Rn (perhaps in Fig S1) informative for reference in the sections evaluating the potential contributions of variations in F_Rn to Rn-derived COS fluxes.

Specific comments:

P6L26: Do the footprints of the flux tower for the EC system overlap with the nearby tall tower? Is it possible that differences arise due to spatial heterogeneity and not any kind of estimation? There could be heterogeneity that affects some gases more than others.

[Figure]

P8L14: Clarify the time period of NEE data using to derive F_Rn

P13L15: Consider citing also Commane et al., Figure 2D

Supplement: Text spacing looks strange
* * *

---

## Author Comment (AC2) · 11 Aug 2017

The comment was uploaded in the form of a supplement:
https://www.atmos-chem-phys-discuss.net/acp-2017-407/acp-2017-407-AC2-supplement.pdf

---

## Author Response (AR1)

We thank the reviewers for their comments and positive feedback. In the following we reply to individual comments of the reviewers and present a few relevant updated figures at the end of this document.

**Referee #1**

In this study, the authors profiled COS and CO2 concentrations at 5 heights for 5 months in 2015 for evaluating storage fluxes and understanding the processes of gas exchange, concomitantly with eddy covariance and radon measurements over and in the canopy for assessing the vertical fluxes. Special attention is paid to the nighttime uptake of COS and to the apportionment of this sink within the ecosystem. I share the conclusions of this paper which is well written and deserves to be published, but more detailed information is required in an area of major importance to the study, i.e. the role of plants in the nighttime uptake of COS which, in this manuscript, is only assessed indirectly (i.e., Plant flux = Total flux – Soil flux) because the authors make very little use of their short-term COS profile measurements. If trees are a larger sink of COS than soils during the night, there should be some sign of COS drawdown at canopy level especially if the 8 m canopy layer is decoupled from the air close to the ground as discussed in chapter 5.2. I look forward seeing new plots showing isolines of the average COS concentration distributions within the canopy as a function of height (including the 125 m reference level) and time of day for the summer and autumn months (see my comment about Fig. 2 below). I hope that this analysis will not end up showing that there are no clear vertical changes in COS between .5 and 125 m height during the night.

In the figures below we show the average concentrations of COS (left) and $CO_2$ (right) per height against time of the day in Jul-Aug (top) and Sep-Nov (bottom). Here we see that during the night a large gradient exists within the canopy for both COS and $CO_2$, of which the largest gradient is close to the surface. Higher in the canopy the gradients are smaller, likely because there exists more turbulent mixing at these canopy levels. When we consider the higher turbulent mixing higher in the canopy, it does not mean that smaller gradients at these higher levels indicate smaller fluxes at the tree foliage compared to the surface. An analysis of gradients within the canopy is therefore not directly indicative for the size of fluxes at different parts within the canopy, unless the concentrations are related to concentrations of another gas with a known flux distribution, like we did using [222]Radon. We agree with the reviewer that we should give the reader an indication of the size of gradients within the canopy within the night, but we do not make further use of gradients in an attempt to quantify fluxes. We added a sentence on the size of gradients in Jul-Aug and Sep-Nov in relation to storage fluxes in section 4.1 (*"The average nighttime gradient between 23 and 0.5 m corresponding to these storage fluxes is 63 ppt for COS and -45 ppm for $CO_2$ (23 – 0.5 m concentration) in Jul-Aug and is 57 ppt and -17 ppm in Sep-Nov."*). We decided to not show the plots (that we show here) in the current manuscript as this would deserve more discussion than only just the nighttime gradients. We also plan to work on a manuscript covering the concentration gradients and changes over time and plan to discuss more details of the gradients there.

[Figure]

Methods

The authors used a multi-position Valco valve to switch frequently (10 times per hour) between the sample tubing from the different profile heights. It would be useful to know the flow rates through the sampling lines (are they flushed permanently or not?) and through the QCLS sampling cell which internal volume could be reminded. Did you use data from the last xx seconds of each cycle or the 3 min records? Did you notice memory effects from previous samples? I would highly recommend the authors to show in a new figure a typical 1h cycle recorded in the late night (stable atmospheric conditions favoring COS and CO2 stratification) and in the afternoon (vertical mixing, no vertical gradient).

We have added the following methodological details in section 2.2.1 to cover these questions:

> The sample tubing was continuously flushed. For the profile measurements, the flow rates were set such that there was a time delay between 30 and 60 s from the moment that the air enters the inlet at different heights until it reaches the cell of the QCLS, which is 17 L min$^{-1}$ for 125 m and 2 L min$^{-1}$ for 4 m. The flow rate from the Valco valve through the sample cell was

*set at 0.15 L min⁻¹ where the sample cell has a volume of 0.5 L. […] The first 60 s of each 3-minute measurement were discarded to account for cell flushing time.*

And in section 2.2.2:

The air is sampled *with a flow of 9 – 10 L min⁻¹* at 23 m height at a small tower that is at 30 m distance from the 125 m tall tower.

As requested by the reviewer, we added two figures in the supplementary material to illustrate typical 1 h cycles in the night with a large concentration gradient and during the day with a small gradient and refer to these figures in section 2.2.1:

[Figure]

Figure S1: A typical 1h cycle of COS and $CO_2$ concentrations during nighttime (01:00 hr) on July 20, 2015, showing the switching between cylinder gases, profile heights (shaded), and soil chambers. A gradient between the different profile heights can be distinguished.

[Figure]

Figure S2: A typical 1h cycle of COS and $CO_2$ concentrations during daytime (14:00 hr) on July 20, 2015, showing the switching between cylinder gases, profile heights (shaded), and soil chambers. A gradient is hardly detectable due to turbulent mixing of the air.

Figures

Fig. 1 provides a nice illustration of the radon-tracer method but the times of sunrise and sunset are missing. I guess that a significant portion of daytime Rn measurements is used to calculate the linear regressions shown in the lower panels from which the nighttime fluxes of COS and CO2 are derived. This appears inconsistent to me. 8 m data extrapolated from other levels using an exponential fit isn't it? I also suggest adding the diurnal variations of hourly values of storage fluxes and friction velocity during 12-13 July 2015.

It is right that part of the afternoon data were included in the analysis and we did not select only the nighttime data because the afternoon data are mostly constant and therefore did not seem to affect the linear regression between the gas concentrations. Nevertheless, we agree that it is more consistent to apply the radon-tracer method for times between sunrise and sunset and have updated the analysis to use only data with sun elevation below 0°. The consequence is that linear regressions between $^{222}$Rn and COS or $CO_2$ are now done with fewer data points, which leads to slightly lower correlations between the gases and a reduced number of nighttime fluxes in the analysis. This analysis is updated throughout the manuscripts ($F_{COS-Rn}$ = -6.8 ±2.2 pmol m$^{-2}$s$^{-1}$) and now leads to slightly lower correlations in Fig 5 (except with u∗), but does not change the overall conclusion of the manuscript. In the relation of $F_{COS-Rn}$ with u∗ we now find a stronger correlation ($R^2$ = 0.33), which does change our conclusion about $F_{COS}$ being dependent on u∗. In section 5.3 we have therefore added the following few sentences:

> *The correlation between nighttime COS or $CO_2$ fluxes and u∗ ($R^2 = 0.50$ for $F_{COS-EC}$ and 0.30 for $F_{CO2-EC}$, not shown) is also observed with the radon-tracer method for $F_{COS-Rn}$ ($R^2 = 0.33$) but not for $F_{CO2-Rn}$ ($R^2 = 0.003$, not shown). This suggests that nighttime COS uptake by plants is limited by the reduced COS concentrations at the leaf boundary layer, which is not the case for $CO_2$. This means that the u∗ filtering that is applied to $F_{COS-EC}$ is possibly an overstated filtering and leads to an overestimated $F_{COS-EC}$, which could explain the difference between $F_{COS-EC}$ and $F_{COS-Rn}$.*

As requested we have also added the storage fluxes and friction velocity in Fig. 1, see Fig 1 at the end of this document.

Fig. 2 shows the mean diurnal variations in fluxes based on all available data with friction velocities > to 0.3 m/s (a quite high threshold to separate stable from turbulent atmospheric conditions). Such a presentation is inadequate because daylight duration from early July to late October exhibits large variations at 61◦N as stated page 3 line 15. I don't think it is necessary to generate monthly averages of hourly fluxes, averaged values for summer and autumn months would be adequate.

We adjusted Fig 2 such that it only represents the summer months (Jul-Aug). In the supplementary material (Fig S4) we added the same figure for the autumn months (Sep-Nov). Also the text is adjusted to indicate both summer and autumn values:

> *The storage fluxes of COS (Fig. 2) are slightly negative during nighttime with an average nighttime value of -0.9 pmol m$^{-2}$ s$^{-1}$ in Jul-Aug and -0.5 pmol m$^{-2}$ s$^{-1}$ in Sep-Nov (Fig. S3). Early in the morning when turbulence is enhanced, the storage fluxes become positive and have an average maximum of 2.1 (1.8) pmol m$^{-2}$ s$^{-1}$ at 09:00 (10:00) in Jul-Aug (Sep-Nov). [...] In this*

dataset, storage fluxes of both COS and $CO_2$ are small compared to the EC flux, i.e. storage fluxes are on average 5 % of $F_{COS-EC}$ and 7 % of $NEE_{EC}$, *with variation between summer and autumn from 4 % (Jul-Aug) to 6 % (Sep-Nov) for $F_{COS-EC}$.*

Abstract

Page 1 line 18: the total nighttime COS fluxes over the whole measurement period were. . .

Corrected as suggested

Page 1 line 21: . . .suggesting that the main sink of COS is not located at the ground. May be the new analysis of vertical profiles will demonstrate that the main sink of COS is not located at the ground.

As explained earlier in our reply, an analysis of the vertical profiles of COS alone would not give further insight into the size of fluxes at different parts within the canopy due to varying turbulent mixing within the canopy. Such an analysis could only be done by relating COS concentrations to other gas concentrations with known flux distributions, which we have done in the best possible way in this paper by relating the COS concentrations to that of [222]Radon.

**Referee #2**

In the article "Canopy uptake dominates nighttime carbonyl sulfide fluxes in a boreal forest" Kooijmans and co-authors present a season of nighttime fluxes of COS and CO2 derived at a height of 8m in a boreal forest with a dominant canopy height of 17 m. Fluxes are derived by eddy covariance, but recognizing the limitations of eddy covariance under placid nighttime conditions, the authors derive fluxes by gradient-flux similarity methods to Radon 222, which is emitted at consistent rates from soil. The authors find evidence for significant nocturnal uptake of COS by the canopy, suggesting a greater role of vegetation than soils in atmospheric COS uptake both during the day and night at this site. The measurement methods and analysis are thorough, and the results provide much needed data to the field including independent measurements of stomatal conductance for comparison with COS fluxes. This is a valuable contribution to understanding the behavior of COS in ecosystems for more precise application as a carbon cycle tracer. General and specific comments follow.

General comments:

The manuscript discusses the possibility that under still conditions, when eddy covariance techniques are not applied due to low u*, COS may be depleted at the leaf surface and slow uptake rates. This is discussed in relation to the suitability of u* filtering. However, a similar phenomenon could occur at the soil-atmosphere interface under still conditions where COS uptake rates are limited by COS availability in depleted layers low in the profile. Under those conditions, emissions of CO2 and 222Rn would however not be limited given that they are production reactions. How would concentration-depletion at the soil-atmosphere interface affect interpretation of the data in this paper (for example the interpretation of Figure 3)?

This is an interesting point. It is true that if the surface air is depleted, the soil uptake of COS can be reduced, whereas the emission of $^{222}$Rn is not limited by a higher concentration above the surface. This can be an extra reason why COS concentrations are less correlated with $^{222}$Rn than $CO_2$. However, Sun et al. (2017) found no correlation between soil COS fluxes and COS concentrations. We added an extra paragraph in section 5.3 to cover this:

> Similar to the limitation on leaf uptake by depleted COS concentrations, soil COS uptake may also be limited by the depleted COS at the soil-atmosphere interface. In contrast, soil emissions of $CO_2$ and $^{222}$Rn do not depend on atmospheric concentrations. This may explain the stronger similarity between $CO_2$ and $^{222}$Rn emissions, which is reflected in the higher correlation between $CO_2$ and $^{222}$Rn concentrations than that between COS and $^{222}$Rn (Fig. 3). However, Sun et al. (2017) found no correlation between soil COS fluxes and COS concentrations ($R^2 < 0.001$) for ambient concentrations between 200 and 450 ppt. This implies that the soil COS flux is not limited by the low ambient concentration at night, and a correlation between $u_*$ and soil COS uptake is not warranted.

It would be useful to discuss the uncertainty in scaling up soil flux measurements from the chamber measurements. How much variation was there between chambers? Given the large difference in footprint between tower-based and chamber measurements, how could spatial heterogeneity affect your estimations of the role of nocturnal canopy uptake of COS?

There could indeed be some heterogeneity in soil fluxes due to the spatial distribution of stones/bedrock and trees, which causes heterogeneity in soil temperature and moisture. The average of the soil fluxes in the two chambers was -2.8 ± 1.0 and -2.5 ± 1.2 pmol m$^2$s$^{-1}$ (Sun et al., 2017). Given the size of the total nighttime flux of around 8 pmol m$^{-2}$ s$^{-1}$ we assume that the average variation of 0.3 pmol m$^{-2}$s$^{-1}$ between the two chambers does not have a substantial effect on the results in this study. We added a sentence on the variability between the two chambers in section 4.2:

> *In comparison, nighttime soil fluxes of COS are on average -2.7 pmol m$^{-2}$ s$^{-1}$ (-2.8± 1.0 and -2.5 ± 1.2 pmol m$^{-2}$ s$^{-1}$ for the two chambers) and soil fluxes do not show a clear diurnal (Fig. 2) or seasonal cycle.*

and in section 5.1:

> *This is also supported by the soil chamber measurements that were on average -2.7 pmol m$^{-2}$ s$^{-1}$ with only little variation between the two chambers,* which suggest that the soil contributes to 34–40 % of the total nighttime COS uptake.

No significant trend of F_Rn derived from NEE was reported with SWC, but was there a trend over the season? I would find a time series of F_Rn (perhaps in Fig S1) informative for reference in the sections evaluating the potential contributions of variations in F_Rn to Rn-derived COS fluxes.

The time series of F$_{Rn}$ have been added to Supplementary figure S1 (now S3, see figure S3 at the end of this document) and a reference is made to this in section 3.2. In section 3.2 we already mentioned that there was no temporal change in F$_{Rn}$:

> *Temporal variation of F$_{Rn}$ can be expected due to the changes in SWC that affects the soil permeability; however, no temporal change or correlation with SWC was found (R$^2$ =0.07) throughout the season (see Fig. S3 in supplementary material).*

 Specific comments:

P6L26: Do the footprints of the flux tower for the EC system overlap with the nearby tall tower? Is it possible that differences arise due to spatial heterogeneity and not any kind of estimation? There could be heterogeneity that affects some gases more than others.

The distance between the two flux towers is only 30 m and the two EC systems are at the same height above the ground. Therefore we do not expect large differences in the footprint and unfortunately we are not able to point to a cause that explains the differences between the fluxes of the two systems.

P8L14: Clarify the time period of NEE data using to derive F_Rn

This is clarified as follows:

> *We derived F$_{Rn}$ over the period from July to October and found an average of* 5.9 mBq m$^{-2}$ s$^{-1}$ with a standard deviation of 3.9 mBq m$^{-2}$ s$^{-1}$ and a standard error of 0.8 mBq m$^{-2}$ s$^{-1}$ (*see Fig. S3 in supplementary material*).

P13L15:  Consider citing also Commane et al., Figure 2D

This reference has been added.

Supplement: Text spacing looks strange

The spacing is now smaller.

**Figures**

[Figure]

Figure 1: COS, $CO_2$ and $^{222}$Rn concentrations, $u_*$ and the storage flux of $CO_2$ ($F_{stor-CO2}$) during 12-13 July 2015 where the data with sun elevation below 0° are used to derive nighttime fluxes of COS and $CO_2$ (black, filled). The bottom figures show the linear regression between $^{222}$Rn and COS (left) and $CO_2$ concentrations (right) on which $F_{COS-Rn}$ and $NEE_{Rn}$ are based.

[Figure]

Figure 2: Correlations of $F_{COS-Rn}$ with $g_{sCOS}$, $T_{air}$, VPD, and $u_*$. All data (except $F_{COS-Rn}$) are averages over individual nights (with nighttime defined as sun elevation below -3°). Data in this plot largely represent a period in August 2015 with dry conditions (i.e. decreasing SWC, and increasing $T_{air}$ and VPD).

[Figure]

Figure S3: Overview of (a) meteorological conditions (SWC, $T_{air}$ and RH), (b) VPD, (c) $g_{sCOS}$, (d) radon-based fluxes $F_{COS-Rn}$ and $NEE_{Rn}$, (e) EC-based fluxes $F_{COS-EC}$ and $NEE_{EC}$ and (f) $F_{Rn}$. 5-day running averages are plotted in corresponding colors. For $g_{sCOS}$, the running average is only plotted up to September 1[st] as only very few data points are available after that period.

[Figure]

Figure S4: Storage fluxes $F_{stor}$ (green), ecosystem fluxes $NEE_{EC}$ and $F_{COS-EC}$ (red) and soil fluxes $F_{soil}$ (blue) of COS (left) and $CO_2$ (right) in autumn (September − November) 2015. Thick lines indicate the median values of the data over the whole measurement period, and the shaded areas specify the $25^{th}$-$75^{th}$ percentiles. The median values of $NEE_{EC}$ and $F_{COS-EC}$ without storage correction are shown in gray. The ecosystem fluxes are filtered for low $u_*$ values with a threshold of 0.3 m s$^{-1}$.